# Emotion Recognition and Self-Reported Emotion Processing in Alcohol and Cannabis Co-Using Young Adults

**DOI:** 10.3390/bs14050407

**Published:** 2024-05-14

**Authors:** Anita Cservenka, Lacey C. Donahue

**Affiliations:** School of Psychological Science, Oregon State University, 2950 SW Jefferson Way, Corvallis, OR 97331, USA; donahlac@oregonstate.edu

**Keywords:** alcohol, cannabis, emotion recognition, socio-emotional processing, alexithymia

## Abstract

Alcohol and cannabis use are each associated with impairments in emotion recognition accuracy, which may promote interpersonal problems. It is unclear if emotion recognition or self-reported emotion processing differs between young adult alcohol and cannabis co-users (ACCs) and healthy controls (HCs). This study examined whether ACCs and HCs differed in their emotion recognition across two different behavioral tasks with static or dynamic faces and determined if there were differences in self-reported socio-emotional processing and alexithymia. 22 ACCs (mean age = 21.27 ± 1.75) and 25 HCs (mean age = 21.48 ± 2.68), matched on age, sex, and IQ, completed the Metrisquare Emotion Recognition Task and CANTAB Emotion Recognition Task. The ACCs and HCs were compared on task accuracy and self-reported measures, including the Social Emotional Questionnaire (SEQ) and the Perth Alexithymia Questionnaire (PAQ). No significant main effects of the Group variable or the Emotion–Group interaction variable were present for either task. The ACCs had lower SEQ (*p* = 0.014) and higher PAQ (*p* = 0.024) scores relative to the HCs, indicating greater difficulties in socio-emotional processing and identifying one’s own emotions, respectively. Understanding the behavioral correlates of the self-reported difficulties in emotion processing reported by ACCs is needed to develop interventions to reduce these symptoms and promote healthy socio-emotional functioning in this population.

## 1. Introduction

### 1.1. Alcohol and Cannabis Co-Use

While alcohol and cannabis use have often been studied as separate substance use behaviors, their co-use is frequently more harmful and leads to greater risk of negative consequences and poorer outcomes than the use of either substance alone [1]. From 2002 to 2018, the prevalence of alcohol and cannabis co-use in both college and non-college students increased [2]. Over the past decade, 13.4% of adolescents have been classified as concurrent alcohol-cannabis users, with over half of high school seniors reporting both lifetime cannabis and alcohol use [3]. Among young adults, cannabis use days are accompanied by greater same-day alcohol use, and elevating the use of one substance can increase the risk for greater use of the other substance [4]. Across the first two years of college, young adults consume more drinks and achieve a higher blood alcohol content on days of cannabis consumption, and increased negative alcohol use consequences are associated with this co-use behavior [5]. Alcohol and cannabis co-use is also associated with more harmful overall consequences, including lower rates of high school graduation, substance use problems in adulthood, increased social repercussions, higher rates of mental health symptoms, and greater involvement in the criminal justice system [1]. Together, these findings indicate that negative consequences related to co-use are highly prevalent in the emerging adult demographic and may present further complications in adulthood. In adulthood, individuals with co-occurring alcohol use disorder (AUD) and cannabis use disorder (CUD) are at higher odds for displaying various psychosocial problems, including difficulty with interpersonal relationships (trouble with employer, colleagues, relatives, friends, and/or partners), relative to individuals with AUD-only or CUD-only diagnoses [6]. These problems in interpersonal relationships may reflect challenges in socio-emotional processing among individuals who use alcohol and/or cannabis. Previous research has frequently investigated socio-emotional processing via tasks that examine the facial emotion recognition accuracy of alcohol or cannabis users relative to healthy controls; understanding whether these impairments are present may help explain difficulties with social relationships that could lead to the maintenance of substance use.

### 1.2. Emotion Recognition and Alcohol Use Disorder

Research suggests that diagnoses of AUD are associated with socio-cognitive impairments [7], such as deficits in emotion recognition [8,9,10,11]. Individuals with AUD not only tend to overestimate the intensity of emotions seen in faces [9,10,11], but they also make more negative emotional attributions to facial expressions [8,11,12]. Additionally, they often confuse one emotion for another, such as mislabeling disgust as anger or contempt [11]. Furthermore, individuals whose primary substance of choice is alcohol, show emotion recognition deficits on angry, disgusted, fearful, and sad faces [13]. These deficits may result in the maintenance of substance use to cope with difficulties in interpersonal relationships that emerge from impaired socio-emotional processing [14]. However, it is less certain if these emotion recognition difficulties are present in heavy binge drinkers (those who engage in episodic drinking in high quantities) without a diagnosed AUD.

### 1.3. Emotion Recognition and Binge Drinking

Few studies have examined the effects of binge drinking (consuming at least four or more drinks per drinking occasion for females, and at least five or more drinks per drinking occasion for males) on emotion recognition in non-clinical populations. One study found that young adult binge drinkers performed worse than the healthy controls in an emotion recognition task and a higher emotional intensity of the faces was required for binge drinkers to perform accurately [15]. A subsequent study found that binge drinkers specifically were less accurate in identifying sadness and fear relative to the healthy controls, but not other emotional expressions. This deficit of identifying fear and sadness was interpreted as impaired functioning of the amygdala and anterior cingulate cortex, respectively [16]. Difficulties in identifying these negative emotions in others could be associated with socio-emotional processing challenges, such as interpersonal problems potentially motivating continued heavy alcohol use. Another study of young binge drinkers suggested that some effects of alcohol use on emotion recognition may be present early in the course of heavy drinking, as high binge-drinking adolescents were less likely to identify a face as fearful relative to low binge-drinking peers [17]. Thus, emotion recognition impairments may be present in community samples of youth and serve as risk factors for the development of substance use disorders. A review of studies examining the effects of binge drinking on several emotional processes, including emotion recognition, indicated that the recognition of fear may be particularly impacted in binge-drinking individuals [18]. However, it is important to consider whether other substance use, such as cannabis use, common during young adulthood, may also contribute to these impairments.

### 1.4. Emotion Recognition and Cannabis Use

Previous research suggests that cannabis use may also be related to atypical socio-emotional processing, but this has not been extensively examined. Abstinent cannabis-dependent individuals were less accurate in the identification of discrete emotional expressions relative to the healthy controls, with findings limited to negative emotional expressions [19]. Another study found an overall group effect, but no interaction with emotion, such that frequent cannabis users had lower accuracy in emotion recognition than the healthy controls for emotions that were displayed using static images, with 20% increments of emotional intensity [20]. In contrast, using a dynamic emotion recognition task in which facial expressions gradually changed in expression, frequent cannabis users had slower emotion recognition abilities relative to the controls but no differences in their overall recognition accuracy [21]. Finally, a recent longitudinal study examining the effect of cannabis use initiation during adolescence on emotion recognition abilities found decreases in emotion recognition accuracy with more frequent cannabis use by late adolescence [22]. These differences across studies could be related to variations in task design (static vs. dynamic facial expressions) and substance use characteristics (abstinent vs. current frequent cannabis users) of the participants included. Thus, it is important for studies to consider the influence of task design on emotion recognition accuracy in alcohol and cannabis co-using young adults. However, to our knowledge, few studies have considered the contributions of alcohol and cannabis use on emotion recognition in the same sample of participants. One study of alcohol- and/or cannabis-using adolescents who were in residential treatment found that the severity of problematic cannabis use was negatively correlated with the accuracy of static emotion recognition for several high-intensity negative emotional expressions, particularly fear and sadness. However, relationships between alcohol use severity and emotion recognition accuracy were not found, suggesting that there may be some substance-specific effects of cannabis in relation to emotion recognition that are present above and beyond co-occurring psychopathologies [23]. Given these findings, it appears relevant to explore whether alcohol and cannabis co-use in young adults is associated with significant differences in their emotion recognition relative to healthy controls and whether the severity of alcohol and cannabis use is related to task performance. Importantly, a review of alcohol and cannabis co-use effects on neurocognitive function indicates mixed findings on the negative outcomes associated with co-use [24]. Previous studies have focused primarily on executive functioning, attention, learning, memory and verbal functioning as opposed to socio-emotional processing, warranting further research in this area.

### 1.5. Alexithymia in Alcohol and Cannabis Users

In addition to impairments in emotion recognition accuracy, individuals with alcohol and substance use disorder have been characterized as having elevated rates of alexithymia [25,26], a deficit in understanding and recognizing one’s own emotions and feelings [27]. Approximately half of individuals with AUD exhibit alexithymic traits [26], with earlier studies indicating even higher prevalence [28]. Alexithymic symptoms have been associated with interpersonal problems, difficulty mentalizing, impairments in theory of mind, empathy, and several psychiatric co-morbidities [29]. In relation to alcohol use, alexithymia has been linked to problematic alcohol use [30,31] and binge drinking [32]. While the presence of alexithymia has been more frequently studied in alcohol users, there is evidence for these symptoms in cannabis users as well. Specifically, adult patients with CUD [33], as well as adolescents and young adults with cannabis use or dependence [34], exhibit greater levels of alexithymia. Greater alexithymia symptoms have also been linked to problematic cannabis use [35]. In another study, adolescents and young adults with cannabis dependence did not have significantly higher overall alexithymia scores relative to the healthy controls, but they had more impairments on a subscale that measured their difficulty in identifying feelings [36]. However, the previous studies have been primarily conducted in clinical populations, and there is limited research on alexithymia symptoms in alcohol and cannabis co-using young adults recruited from the community. Given the prevalence of alcohol and cannabis co-use among young adults in the general population, it is important to consider whether difficulties in recognizing and identifying one’s emotions are correlated with substance use characteristics. Thus, a secondary aim of this study was to examine group differences in self-reported socio-emotional functioning and alexithymia symptoms in alcohol and cannabis co-users (ACCs) relative to healthy controls (HCs).

### 1.6. Current Study

To our knowledge, no studies have investigated the effects of binge drinking with co-occurring cannabis use on emotion recognition or alexithymia symptoms in young adults. Since affect recognition is critical to healthy socio-emotional functioning, it is essential to understand how frequent use of cannabis and alcohol is associated with emotion processing in young adults. Thus, the goals of the current study were (1) to determine whether there are group differences in emotion recognition accuracy between binge drinkers with co-occurring cannabis use and healthy controls, and (2) to examine if these differences were task-dependent. We chose two emotion recognition tasks that differed in the static vs. dynamic (expression intensifying during the course of presentation) nature of the emotional faces presented. The purpose of utilizing two tasks was to determine if group differences in performance could be explained by the method in which faces are presented, as dynamic faces may be more realistic [37] and have been associated with greater recognition accuracy compared to static faces [38]. Additionally, the tasks were selected to be comparable on other features (e.g., the task length and number of trials) to avoid differences that would influence the interpretation of findings. We chose to focus on emotion recognition accuracy (as opposed to identification bias, reaction time, etc.), as this is the most commonly examined dependent variable across the previous literature, and we intended to limit the number of tests to avoid Type I errors. Based on prior research, we hypothesized that the ACCs would have poorer emotion recognition accuracy relative to the HCs and that this would be present in emotions depicting negatively valenced facial expressions (e.g., angry, sad, fearful, and disgusted) in a task utilizing static faces, as seen across previous studies with participants diagnosed with AUD or CUD. We also hypothesized that self-reported alexithymia symptoms would be elevated, while self-reported socio-emotional functioning would be poorer in the ACCs relative to the HCs.

## 2. Methods

### 2.1. Participant Recruitment

Participants were recruited through our existing database of previous research participants who agreed to be contacted about future studies, and flyers posted on college campuses, in cannabis dispensaries, coffee shops, community centers, social media websites, and Craigslist.

### 2.2. Inclusionary and Exclusionary Criteria

Following informed consent, eligibility interviews were conducted in person or by phone with each participant to determine their study eligibility. The inclusionary criteria for the ACC group were (1) ≥3 days of cannabis use/week in the past year, (2) use of cannabis in the past month, and (3) ≥6 binge drinking episodes in the past six months (≥4/5 drinks per drinking occasion for female/male participants [39]. The inclusionary criteria for the HC group was (1) ≤10 lifetime uses of cannabis, (2) no cannabis use in the past month, and (3) no binge drinking history. Given the high prevalence of any lifetime cannabis or alcohol use by young adulthood [40], we allowed low levels of cannabis and alcohol use in the current control sample to enhance the ecological validity and generalizability of the study’s findings. Limited cannabis use history was permissible in the HC group, as reported for the other healthy controls [19,20,21]. Given that over 67% of young adults report past month alcohol use [40], we did not exclude HCs for lifetime or past month alcohol use. Since there have been previously reported effects of binge drinking on emotion processing (e.g., [15]), we excluded HCs for any history of binge drinking.

The exclusionary criteria for all participants included: age <18 or >30; lifetime history of a self-reported psychiatric disorder; current use of psychotropic medications; major neurological/medical illness; self-reported learning disability; uncorrected vision impairments; head injury with loss of consciousness >2 min; and prenatal exposure to drugs or alcohol. Furthermore, the exclusionary criteria for the ACC group included >15 lifetime uses of illicit substances combined (other than cannabis). The exclusionary criteria for the HC group included any lifetime illicit substance use and ≥90 lifetime days of cigarette use. These exclusionary criteria were set forth to remove participants who may have emotion recognition performance related to confounding variables that were not of interest for this study.

### 2.3. Study Procedures

Eligible participants were invited to the laboratory to complete a study visit consisting of a breathalyzer to ascertain the absence of acute alcohol intoxication, the collection of a urine sample to determine recent substance use, questionnaires assessing demographics, substance use, and emotion processing, and two emotion recognition tasks, which were completed in the same order by all participants, in line with other research utilizing two emotion processing tasks in a between-groups design [17]. The two emotion recognition tasks did not directly follow each other but were separated by several questionnaires that the participants completed (not reported in the current study). Given the distinct nature of the facial stimuli used in each task, and no within-subjects aims of the current study, carryover effects were not expected. The participants were asked to abstain from all substance use for ≥12 h prior to the study visit to prevent the effects of acute intoxication on the completion of study measures (all participants had readings of 0.00 mg/dL on the breathalyzer at the start of their study visits). The 12-panel urinary panel (CLIA Waived, Inc., Carlsbad, CA, USA) tested for tetrahydrocannabinol (THC), cocaine, opiates, amphetamine, methamphetamine, phencyclidine, barbiturates, benzodiazepenes, methadone, methylenedioxymethamphetamine (MDMA), oxycodone, and buprenorphine. Informed consent was obtained from all participants prior to engaging in the study activities. Following their visit, the participants were compensated with an Amazon e-gift card. This study was conducted according to the guidelines of the Declaration of Helsinki and approved by the Institutional Review Board of Oregon State University (Study #8845, approved 7 December 2018). Additionally, to protect participant confidentiality, a Certificate of Confidentiality was obtained from the National Institutes of Health.

### 2.4. Substance Use Measures

The participants completed the Cannabis Use Disorder Identification Test-Revised (CUDIT-R; [41]), and Alcohol Use Disorder Identification Test (AUDIT; [42]). Scores of eight or higher on the CUDIT-R and AUDIT are indicative of hazardous cannabis use and alcohol use, respectively. Scores of 12 or higher on the CUDIT-R are indicative of a possible CUD, while scores of 13 or higher for females and 15 or higher for males on the AUDIT are indicative of possible AUD. As the HCs did not report cannabis use in the past six months, they did not complete the CUDIT-R. 

### 2.5. Measure of General Intelligence

The two-subtest version of the Wechsler Abbreviated Scale of Intelligence-II (WASI-II; [43]) was administered to all participants to calculate their verbal and matrix reasoning scores and full-scale IQ.

### 2.6. Emotion Recognition Tasks

#### 2.6.1. Metrisquare Emotion Recognition Task (M-ERT)

The M-ERT [44,45] was used as a measure of dynamic emotion recognition and was the first emotion recognition task completed by the participants. The M-ERT uses morphed video clips of faces, with emotion intensities varying between 40 and 100% in 20% increments. A total of 96 trials are presented (16 for each of the 6 emotions: happy, sad, anger, disgust, surprise, and fear), with the task lasting approximately 10 min, completed on a desktop computer. The dynamic faces change from a neutral expression to one of the six emotions for 1–3 s and remain on the screen during the emotion selection. The M-ERT displays the emotion of the face until the participant makes a response.

#### 2.6.2. CANTAB Emotion Recognition Task (C-ERT)

Following the M-ERT and subsequent questionnaires (not reported in the current study), the participants completed the C-ERT (Cambridge Cognition Ltd., Cambridge, UK) on an iPad. The C-ERT requires participants to correctly identify emotions of computer-morphed static facial expressions (happiness, sadness, anger, disgust, surprise, and fear) of real individuals in a computerized task of 90 assessed trials lasting approximately 9 min. Each face appears for 200 ms and ranges from intensity of 1 to 15, followed by a mask lasting 250 ms, after which participants make a choice between the six emotion options. The task has been used in the emotion recognition and substance use literature [12] to examine the effects of AUD on socio-emotional processing. 

### 2.7. Self-Report Measures of Emotion Processing

The Social Emotional Questionnaire (SEQ; [46,47]) was completed by the participants to determine their self-reported socio-emotional functioning. This questionnaire includes 30 items that contribute to five factors: emotion recognition, empathy, social conformity, antisocial behavior, and sociability. Participants rate each item on a scale of 1–5 to indicate how much they agree with the statement, with “1” corresponding to strongly disagree and “5” corresponding to strongly agree (four of the items are reverse-scored). An example item from the emotion recognition factor is: “I notice when other people are frightened.”; an example item from the empathy factor is: “When others are sad, I comfort them.”; an example item from the social conformity factor is: “I co-operate with others.”; an example item from the antisocial behavior factor is: “I avoid arguments.”; and an example item from the sociability factor is: “I am sociable.” Lower scores on the SEQ are indicative of poorer socio-emotional functioning. The total score on the SEQ was used as the dependent variable for analysis. Cronbach’s α was 0.77, indicating acceptable reliability, in line with previous research [46].

The Perth Alexithymia Questionnaire (PAQ; [48]) was completed by the participants as a measure of their ability to self-identify their emotions. The PAQ consists of 24 items that contribute to five subscales and five composite scores. Participants rate each item on a scale of 1–5, with “1” indicating strongly disagree and “5” indicating strongly agree. The total score of the PAQ measures “overall alexithymia; difficulty focusing attention on and appraising one’s own feelings”. Thus, higher PAQ total scores reflect greater alexithymia symptoms. The total score of the PAQ was used as the dependent variable for analysis. The Cronbach’s α was 0.92, indicating excellent reliability.

### 2.8. Data Analysis

Total accuracy on the M-ERT and C-ERT were compared between the participant groups using independent samples *t*-tests. A mixed-model analysis of variance (ANOVA), with “Emotion” as the within-subjects variable and “Group” as the between-subjects variable, was used to examine the main effects of Emotion (six levels) and Group (two levels), as well as the Emotion–Group interaction for recognition accuracy on each task. Furthermore, independent samples *t*-tests were used to examine group differences in the SEQ and PAQ total scores. Follow-up Pearson’s correlations were used to determine the relationships among self-reported socio-emotional functioning and alexithymia (via the SEQ and PAQ, respectively), AUDIT, and CUDIT-R scores in the ACC group. Additional Pearson’s correlations examined the associations between the total accuracy scores for emotion recognition (via the M-ERT and C-ERT), AUDIT, and CUDIT-R scores in the ACC group. Finally, to determine the relationship between emotion recognition performance and self-reported emotion processing, Pearson’s correlations were conducted between the M-ERT/C-ERT total accuracy scores and the SEQ/PAQ total scores across the entire sample.

## 3. Results

### 3.1. Demographics

A total of 22 ACCs and 25 HCs were eligible for the current study and completed all study procedures (Table 1). To summarize, the ACCs and HCs did not significantly differ in age, sex ratio, or WASI-2 subtest IQ (all *p*’s > 0.05). As expected, the ACCs had significantly greater AUDIT scores relative to the HCs (*p* < 0.001). The mean CUDIT-R scores for the ACCs indicated possible CUD on average in the current sample. A total of 19 ACCs tested positive for THC, while 3 ACCs tested negative for THC. All HCs tested negative for THC and all other substances. Three ACCs tested positive for amphetamine.

### 3.2. Emotion Recognition Tasks

Table 2 displays the total accuracies for the ACCs and HCs on the M-ERT and C-ERT. No significant group differences were present for total accuracy on either task (M-ERT: *p* = 0.26; C-ERT: *p* = 0.74). Figure 1A,B present the M-ERT and C-ERT accuracies, respectively, for each Emotion, divided by Group. For the M-ERT, the assumption of sphericity was violated (*W* = 0.48, Χ^2^(14) = 31.29, *p* = 0.005); thus, the Huynh–Feldt correction was used to correct for degrees of freedom (ϵ = 0.98). The ANOVA indicated no significant main effect of Group (*F*(1, 45) = 1.31, *p* = 0.26. partial η^2^ = 0.03), but a significant main effect of Emotion (*F*(4.91, 220.89) = 112.83, *p* < 0.001, partial η^2^ = 0.72). Pairwise-Bonferroni-corrected (*p* < 0.05) comparisons indicated that anger was recognized significantly better than disgust, fear, sadness, and surprise, but significantly worse than happiness (*p*’s < 0.001). Disgust was recognized significantly better than fear (*p* < 0.001), sadness (*p* < 0.001), and surprise (*p* = 0.003), but significantly worse than happiness (*p* < 0.001). Fear was recognized significantly worse than happiness, sadness, and surprise (*p*’s < 0.001). Happiness was recognized significantly better than sadness and surprise (*p*’s < 0.001). There were no significant differences between recognizing sadness and surprise (*p* = 1.00). There was no significant Emotion–Group interaction (*F*(4.91, 220.89) = 1.77, *p* = 0.12, partial η^2^ = 0.04).

For the C-ERT, the assumption of sphericity was violated (*W* = 0.50, Χ^2^(14) = 30.34, *p* = 0.007); thus, the Huynh–Feldt correction was used to correct for degrees of freedom (ϵ = 0.92). The ANOVA indicated no significant main effect of Group (*F*(1, 45) = 0.11, *p* = 0.75. partial η^2^ = 0.002) but a significant main effect of Emotion (*F*(4.61, 207.43) = 53.17, *p* < 0.001, partial η^2^ = 0.54). Pairwise-Bonferroni-corrected (*p* < 0.05) comparisons indicated that anger was recognized significantly worse than disgust, happiness, sadness, and surprise (*p*’s < 0.001) but not significantly different than fear (*p* = 1.00). Disgust was recognized significantly better than fear (*p* < 0.001) and significantly worse than happiness (*p* < 0.001), but not significantly differently from sadness or surprise (*p* = 1.00). Fear was recognized significantly worse than happiness, sadness, and surprise (*p*’s < 0.001). Happiness was recognized significantly better than sadness and surprise (*p*’s < 0.001). There were no significant differences between recognizing sadness and surprise (*p* = 1.00). There was no significant Emotion–Group interaction (*F*(4.61, 207.43) = 0.91, *p* = 0.47, partial η^2^ = 0.02).

### 3.3. Self-Reported Measures of Emotion

Table 3 shows the total SEQ and PAQ scores for the ACCs and HCs. The independent samples *t*-test indicated significantly lower SEQ scores for the ACCs relative to the HCs (*p* = 0.01), suggesting greater self-reported difficulties with socio-emotional functioning. The independent samples *t*-test indicated significantly higher PAQ scores for the ACCs relative to the HCs (*p* = 0.02), suggesting greater alexithymia symptoms in the ACCs relative to the HCs. 

### 3.4. Associations among Task Performance, Self-Report, and Substance Use Severity Measures

Higher SEQ scores were marginally correlated with lower CUDIT-R scores (*r* = −0.38, *p* = 0.09, uncorrected), indicating that better self-reported socio-emotional functioning is related to less problematic cannabis use in ACCs. There were no other correlations between the SEQ and AUDIT or the PAQ and AUDIT or CUDIT-R scores (all *p*’s > 0.10). There were no significant correlations between the total accuracy on the M-ERT or C-ERT and CUDIT-R or AUDIT scores among the ACCs (all *p*’s > 0.30). Across the entire participant sample, there was a trend for a positive correlation between the total accuracy in the C-ERT and self-reported alexithymia (*r* = 0.26, *p* = 0.07, uncorrected), indicating that better emotion recognition accuracy is related to greater difficulties identifying one’s own emotions. There were no other correlations between task performance on the M-ERT and PAQ or SEQ scores or the C-ERT and SEQ scores (all *p*’s > 0.30).

## 4. Discussion

The current study examined whether a common pattern of cannabis and alcohol co-use during young adulthood is associated with impairments in emotion recognition in a non-clinical population. Two different emotion recognition tasks were administered to participants to determine whether task parameters may contribute to performance differences between ACCs and HCs. Additionally, we examined whether perceptions of self-reported socio-emotional functioning or alexithymia differed between the ACCs relative to the HCs. While no Group or Emotion–Group interaction was present in the emotion recognition tasks, the ACCs self-reported more socio-emotional processing difficulties and greater symptoms of alexithymia relative to the HCs.

The current findings differ from previous studies on participants with AUD and CUD, in which clinical populations showed significantly poorer emotion recognition accuracy relative to healthy controls [7,19]. However, in the current study, we examined emotion recognition in young adults with frequent alcohol and cannabis use who were recruited from the community and may have less severe substance use characteristics and problems relative to participants with current or former AUD and CUD diagnoses. Thus, it is possible that emotion recognition differences were not detected because substance use problems have not yet progressed to the same degree in the participants of the current study. In support of this interpretation, greater intensity of binge drinking is associated with differences in emotion recognition compared to lower-intensity binge drinking [17], indicating that even within a community sample, the frequency and/or severity of substance use may be an important determinant of emotion recognition performance. Given the lack of studies examining socio-emotional processing in alcohol and cannabis co-users relative to healthy controls, further research in this area is warranted [24].

While both emotion recognition tasks showed a main effect of Emotion, the pairwise comparisons indicated differences in recognition accuracy by emotional expression, suggesting that the task parameters may affect emotion recognition accuracy in young adults. Specifically, the emotion of anger was recognized worse than four other emotions in the C-ERT, but better than four other emotions in the M-ERT. Additionally, the disgust accuracy scores were similar to the sadness and surprise scores in the C-ERT, but better than the sadness and surprise scores in the M-ERT. These findings support a study on healthy participants who completed different emotion recognition tasks (static faces, dynamic faces, and videos), where the results indicated that emotion accuracy differences are task-dependent [49]. Descriptive statistics from the M-ERT used in the previous study are in alignment with the current findings, indicating that participants were more accurate in identifying angry faces in the M-ERT compared to disgust, fear, sadness, and surprise, and more accurate regarding disgust compared to sadness and surprise; however, the same results were not found for an emotion recognition task that used the static Ekman faces [49]. This indicates that emotion recognition task selection may be a critical factor contributing to group differences or emotion–group interactions for alcohol and/or cannabis users and healthy controls in previous studies.

Although task performance differences were not found between the ACCs and HCs, this study supports and expands upon prior research that reported individuals with AUD and CUD have greater symptoms of alexithymia relative to healthy controls [26,33]. In our study, young adult ACCs reported more symptoms of alexithymia, which adds to the growing literature indicating that these symptoms are also elevated in non-clinical populations and may be an important target for intervention to prevent long-term impairments in emotion processing and associated negative health outcomes. The current study also expands upon previous research by implementing the SEQ in a substance-using population, as ACCs reported more problems with socio-emotional functioning (emotion recognition, empathy, social conformity, antisocial behavior, and sociability) via this questionnaire. Given the self-reported difficulties in emotion processing across both the SEQ and PAQ, this study provides further support to indicate that ACCs may have difficulties with emotional functioning, which are identifiable even in a relatively modest sample size of participants. Interestingly, we found no significant group differences in emotion recognition accuracy, yet both self-reported socio-emotional difficulties and alexithymia were significantly elevated in the ACCs relative to the HCs. Additionally, across the entire sample of participants no significant correlations between task performance and self-reported emotion processing were observed. However, this discrepancy between behavioral performance and self-reported findings may not be unusual. Researchers have argued that the higher reliability of self-reported vs. performance-based measures (prone to greater error variance), more general responses in self-reported vs. more specific responses provided in behavioral tasks, and one-time sampling of these variables [50,51] could explain the lack of correlation among these measures. Even in the absence of significant performance between the groups, self-reported difficulties in emotion processing could indicate a lower likelihood of seeking social support and a greater chance of maintaining substance use to cope with negative affect [25]. Further work is needed to understand the behavioral correlates of self-reported challenges in emotion processing, whether they precede or follow frequent substance use and their effects on interpersonal relationships in ACCs.

### Strengths and Limitations

To our knowledge, this is the first study to examine emotion processing (both behavioral performance and self-reported) in young adult ACCs relative to HCs. Previous studies have largely focused on adults with current or previous AUD or alcohol dependence diagnoses, or adults with current or previous CUD or cannabis dependence diagnoses. Given the high prevalence of alcohol and cannabis co-use during young adulthood, this study contributes to the examination of common patterns of substance use to advance the ecological validity of previous research. Furthermore, the use of two different emotion recognition tasks that varied in the length of their stimulus presentation and the static or dynamic nature of the facial expressions, provided multiple ways to assess whether the mixed findings across prior studies may have been related to differences in task design. While the two emotion recognition tasks were presented in a fixed order in the current study, future research using a within-subjects design should counterbalance task presentation between participants and across appointments in order to minimize any carryover effects. Additionally, the exclusionary criteria were used to limit the study participants to individuals who did not have co-morbidities of lifetime psychiatric diagnoses, neurological disorders, current use of psychotropic medications, or learning disabilities. Thus, a careful attempt was made to prevent the presence of confounding variables that would challenge the interpretation of findings. Finally, this study included self-reported measures of emotion processing largely absent from the previous literature that only used behavioral indices of emotion recognition.

While the current study included several strengths mentioned above, its limitations should be acknowledged. Although the total sample size (N = 47) was relatively small, it is in line with the sample sizes seen across previous studies comparing alcohol or cannabis users with healthy controls (total N = 28–94, mean N = 53.6; [8,9,10,11,12,15,16,19,20,21]), providing an appropriate comparison to earlier research. Future studies should attempt to increase the reliability and external validity of the current findings by recruiting larger sample sizes, including clinical populations, and employing longitudinal designs [22] to determine whether socio-emotional impairments may be a risk factor and/or consequence of frequent alcohol and/or cannabis use during adolescence and young adulthood. The Adolescent Brain Cognitive Development Study will be useful in providing both a large sample size and multiple time points to examine this question [52,53]. While initial attempts to recruit a sample of binge drinking young adults with minimal history of cannabis use (<10 lifetime occasions) was conducted, we were unsuccessful in recruiting such as a sample. This may largely be due to the location of recruitment within the Pacific Northwest United States, in a state where recreational use of cannabis is legal. Recent studies have reported both increases in cannabis use and decreases in alcohol use in states with legal recreational cannabis use [54], suggesting possible substitution effects and reflecting a changing landscape of young adult substance use within the United States. Other studies have indicated that recreational cannabis legalization (RCL) has been associated with a greater likelihood of alcohol and cannabis co-use in adolescents [55], supporting complementarity substance use patterns, which may explain the greater ease of recruiting this sample in the current geographic location. However, mixed findings across studies suggest that the long-term impacts of RCL on the co-use of cannabis and alcohol are not yet known [56]. Finally, it should be noted that the difficulty in recruiting a frequent alcohol-only comparison group precludes the understanding of substance-specific effects when comparing ACCs to HCs but also highlights the potential for greater ecological validity of a co-use sample. It is unclear whether the current lack of significant behavioral differences is related to the recruitment of a non-clinical sample (with less substance use frequency, history, and/or problems) or the co-use of alcohol and cannabis. It is possible that alcohol and cannabis co-use may have somewhat opposing effects on brain and behavior that could counteract each other (as has been suggested in other studies, e.g., [57,58]) and contributed to the lack of group differences in behavioral performance. Thus, future work is needed to expand studies on the effects of alcohol and cannabis co-use on socio-emotional processing to both clinical and non-clinical populations.

## 5. Conclusions

In the current study, we found that young adults characterized by a pattern of frequent binge drinking and cannabis use self-reported more socio-emotional difficulties and greater symptoms of alexithymia relative to healthy controls but did not display differences in emotion recognition task accuracy. Future studies should attempt to determine the behavioral correlates of these self-reported emotion processing challenges to prevent the development of long-term problems related to frequent alcohol and cannabis use during young adulthood.

## Figures and Tables

**Figure 1 behavsci-14-00407-f001:**
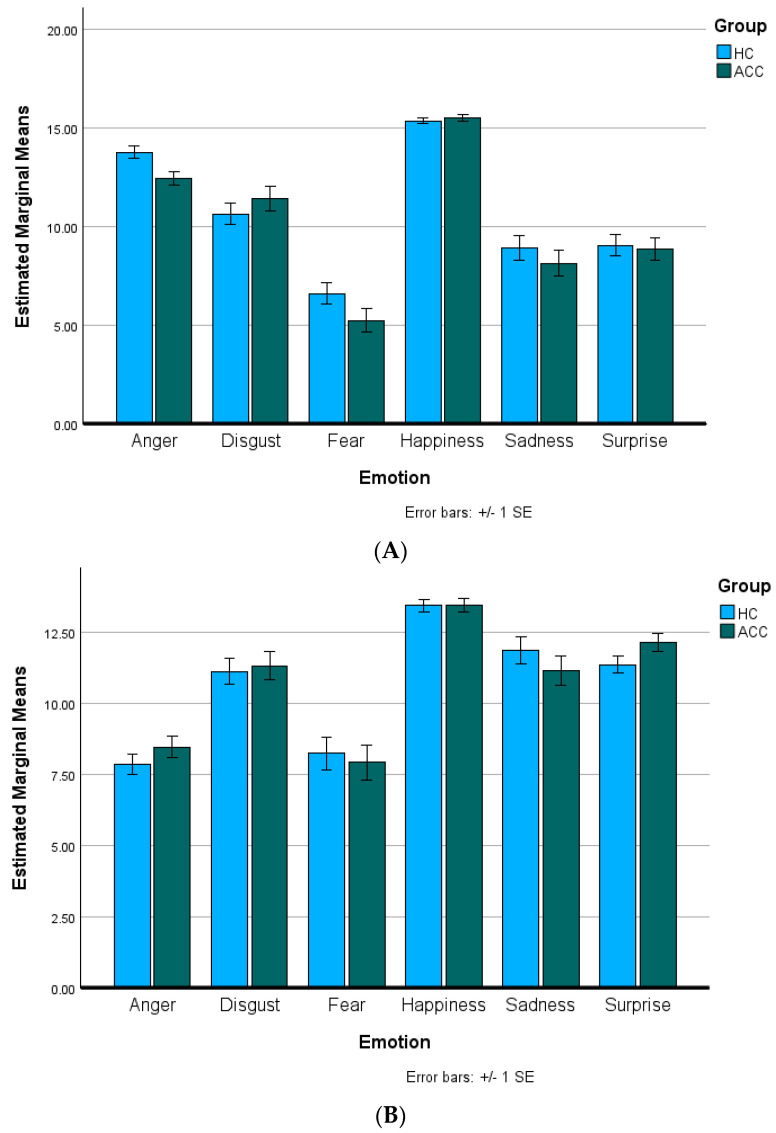
Emotion recognition task accuracy. Total accuracy divided by Emotion and Group for alcohol and cannabis co-users (ACCs) and healthy controls (HCs) on the (**A**) Metrisquare Emotion Recognition Task and (**B**) CANTAB Emotion Recognition Task. There were no significant main effects of Group or Emotion–Group interactions for either task (all *p*’s > 0.05).

**Table 1 behavsci-14-00407-t001:** Demographic characteristics.

	ACC (N = 22)	HC (N = 25)	*t* or *Χ*	*p*
Age	21.27 (1.75)	21.48 (2.68)	0.31	0.76
Sex	17 M/5 F	17 M/8 F	0.50	0.48
WASI ^a^-2 subtest IQ	112.36 (11.62)	114.38 (9.52) ^b^	0.64	0.52
AUDIT ^c^	8.68 (3.06)	1.50 (1.04) ^d^	−10.30	<0.001
CUDIT-R ^e^	13.95 (4.98)	-		

^a^ Wechsler Abbreviated Scale of Intelligence; ^b^ N = 1 missing for HC; ^c^ Alcohol Use Disorders Identification Test; ^d^ N = 18 for HCs because 7 participants had no lifetime alcohol use; ^e^ Cannabis Use Disorder Identification Test-Revised; mean (SD), unless otherwise noted.

**Table 2 behavsci-14-00407-t002:** Emotion recognition task accuracy total scores.

	ACC (N = 22)	HC (N = 25)	*t*	*p*
M-ERT ^a^ Total Accuracy	61.55 (7.61)	63.32 (8.83)	1.15	0.26
C-ERT ^b^ Total Accuracy	64.41 (4.10)	63.84 (7.23)	−0.34	0.74

^a^ Metrisquare Emotion Recognition Task. ^b^ CANTAB Emotion Recognition Task. Mean (SD).

**Table 3 behavsci-14-00407-t003:** Socio-emotional processing and alexithymia scores.

	ACC (N = 22)	HC (N = 25)	*t*	*p*	Cohen’s *d*
SEQ ^ab^	79.05 (20.36)	91.32 (6.32)	2.66	0.01	0.85
PAQ ^c^	78.06 (21.49)	63.16 (22.10)	−2.33	0.02	−0.68

^a^ Social Emotional Questionnaire; ^b^ N = 1 missing for ACC. ^c^ Perth Alexithymia Questionnaire. Lower scores on the SEQ represent greater difficulties with socio-emotional processing. Higher scores on the PAQ represent greater alexithymia symptoms. Mean (SD).

## Data Availability

The raw data supporting the conclusions of this article will be made available by the authors upon request.

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
