# Peer review of "Emotion Recognition and Self-Reported Emotion Processing in Alcohol and Cannabis Co-Using Young Adults"

_behavsci, 2024, doi:10.3390/bs14050407_

Round 1
Reviewer 1 Report
Comments and Suggestions for Authors
The study "Emotion Recognition and Socio-Emotional Processing in Young Adult Alcohol and Cannabis Co-Users" sheds light on an important yet understudied aspect of emotion processing in young adults who are alcohol and cannabis co-users (ACC). The paper explores whether ACC individuals exhibit differences in emotion recognition and self-reported socio-emotional processing compared to healthy controls (HC).
The study's findings offer valuable insights into the socio-emotional difficulties experienced by young adult alcohol and cannabis co-users (ACC). However, a notable limitation is the relatively small sample size utilized in the research. With only 22 ACC and 25 healthy control (HC) participants, the study may lack the statistical power necessary to draw robust conclusions. Though the authors mentioned in their discussion that some of past studies used small sample sizes (e.g., 28-94), the current sample sizes are still smaller than lower bond of the range and mean. Moreover, the limited sample size raises questions regarding the generalizability of the findings to broader populations of ACC individuals. Future research endeavors could consider expanding the sample size and from clinical sites or employing longitudinal designs. By doing so, researchers could enhance the reliability and external validity of their findings, thus contributing to a more comprehensive understanding of emotion recognition and socio-emotional processing in this population.
Another area that warrants attention is the lack of a thorough review of current literature on emotion recognition impairments in alcohol and cannabis users. While the study addresses an important topic, a more comprehensive review of existing research could strengthen its theoretical framework and contextualize the findings within the broader literature. A comprehensive literature review would not only help identify gaps or contradictions in the existing research but also inform hypotheses, methodology, and interpretation of results. By synthesizing previous findings, researchers could provide readers with a more nuanced understanding of the factors influencing emotion recognition and socio-emotional processing in ACC individuals. Some papers that have been overlooked:
Bedillion, M. F., Blaine, S. K., Claus, E. D., & Ansell, E. B. (2021). The effects of alcohol and cannabis co-use on neurocognitive function, brain structure, and brain function. Current behavioral neuroscience reports, 1-16.
Blair, R. J. R., Bashford-Largo, J., Zhang, R., Mathur, A., Schwartz, A., Elowsky, J., ... & Blair, K. S. (2021). Alcohol and cannabis use disorder symptom severity, conduct disorder, and callous-unemotional traits and impairment in expression recognition. Frontiers in psychiatry, 12, 714189.
Leganes-Fonteneau, M., Pi-Ruano, M., & Tejero, P. (2020). Early signs of emotional recognition deficits in adolescent high-binge drinkers. Substance Use & Misuse, 55(2), 218-229.
Lannoy, S., Duka, T., Carbia, C., Billieux, J., Fontesse, S., Dormal, V., ... & Maurage, P. (2021). Emotional processes in binge drinking: A systematic review and perspective. Clinical psychology review, 84, 101971.
Furthermore, the rationale behind the selection of two different emotion recognition tasks is not clearly articulated in the paper. While the study utilizes the CAN-TAB Emotion Recognition Task and the Metrisquare Emotion Recognition Task, it does not justify why these specific tasks were chosen or how they complement each other. Providing a rationale for task selection would enhance the transparency and rigor of the study's methodology. For instance, researchers could explain whether the tasks assess different aspects of emotion recognition, such as static vs video, and how they contribute to a comprehensive assessment of socio-emotional processing in ACC individuals. Additionally, discussing any advantages or limitations of each task would help readers understand the rationale behind their inclusion in the study. By addressing these points, researchers could strengthen the methodological rigor of their study and provide a more comprehensive analysis of emotion recognition and socio-emotional processing in young adult ACC individuals.
Author Response
The study "Emotion Recognition and Socio-Emotional Processing in Young Adult Alcohol and Cannabis Co-Users" sheds light on an important yet understudied aspect of emotion processing in young adults who are alcohol and cannabis co-users (ACC). The paper explores whether ACC individuals exhibit differences in emotion recognition and self-reported socio-emotional processing compared to healthy controls (HC).
- The study's findings offer valuable insights into the socio-emotional difficulties experienced by young adult alcohol and cannabis co-users (ACC). However, a notable limitation is the relatively small sample size utilized in the research. With only 22 ACC and 25 healthy control (HC) participants, the study may lack the statistical power necessary to draw robust conclusions. Though the authors mentioned in their discussion that some of past studies used small sample sizes (e.g., 28-94), the current sample sizes are still smaller than lower bond of the range and mean. Moreover, the limited sample size raises questions regarding the generalizability of the findings to broader populations of ACC individuals. Future research endeavors could consider expanding the sample size and from clinical sites or employing longitudinal designs. By doing so, researchers could enhance the reliability and external validity of their findings, thus contributing to a more comprehensive understanding of emotion recognition and socio-emotional processing in this population.
We agree with the reviewer that a limitation of the current study is the smaller sample size. The total sample size is 47 participants, and the values referenced in the Discussion also refer to previous total sample sizes of studies (mean = 53.6). We have now updated these citations with an additional study (also referenced in the Methods section) in which AUD and healthy control groups were compared (Freeman et al., 2018). Thus, the current study’s total sample size is within the range of several previous publications examining emotion recognition task accuracy for which group comparisons have been examined with the inclusion of a healthy control group. We have also indicated the need for larger samples (including clinical populations) using longitudinal designs to enhance reliability and external validity.
- Another area that warrants attention is the lack of a thorough review of current literature on emotion recognition impairments in alcohol and cannabis users. While the study addresses an important topic, a more comprehensive review of existing research could strengthen its theoretical framework and contextualize the findings within the broader literature. A comprehensive literature review would not only help identify gaps or contradictions in the existing research but also inform hypotheses, methodology, and interpretation of results. By synthesizing previous findings, researchers could provide readers with a more nuanced understanding of the factors influencing emotion recognition and socio-emotional processing in ACC individuals. Some papers that have been overlooked:
Bedillion, M. F., Blaine, S. K., Claus, E. D., & Ansell, E. B. (2021). The effects of alcohol and cannabis co-use on neurocognitive function, brain structure, and brain function. Current behavioral neuroscience reports, 1-16.
Blair, R. J. R., Bashford-Largo, J., Zhang, R., Mathur, A., Schwartz, A., Elowsky, J., ... & Blair, K. S. (2021). Alcohol and cannabis use disorder symptom severity, conduct disorder, and callous-unemotional traits and impairment in expression recognition. Frontiers in psychiatry, 12, 714189.
Leganes-Fonteneau, M., Pi-Ruano, M., & Tejero, P. (2020). Early signs of emotional recognition deficits in adolescent high-binge drinkers. Substance Use & Misuse, 55(2), 218-229.
Lannoy, S., Duka, T., Carbia, C., Billieux, J., Fontesse, S., Dormal, V., ... & Maurage, P. (2021). Emotional processes in binge drinking: A systematic review and perspective. Clinical psychology review, 84, 101971.
We appreciate the reviewer’s recommendations for integrating these additional studies on alcohol and cannabis use and emotion recognition and agree that they are foundational to the aims of the current study. We have now included them in the Introduction and have incorporated relevant studies when interpreting the current findings in the Discussion section.
- Furthermore, the rationale behind the selection of two different emotion recognition tasks is not clearly articulated in the paper. While the study utilizes the CAN-TAB Emotion Recognition Task and the Metrisquare Emotion Recognition Task, it does not justify why these specific tasks were chosen or how they complement each other. Providing a rationale for task selection would enhance the transparency and rigor of the study's methodology. For instance, researchers could explain whether the tasks assess different aspects of emotion recognition, such as static vs video, and how they contribute to a comprehensive assessment of socio-emotional processing in ACC individuals. Additionally, discussing any advantages or limitations of each task would help readers understand the rationale behind their inclusion in the study. By addressing these points, researchers could strengthen the methodological rigor of their study and provide a more comprehensive analysis of emotion recognition and socio-emotional processing in young adult ACC individuals.
We agree that the justification for selecting the two emotion recognition tasks to compare performance requires elaboration in the manuscript. The purpose of selecting these tasks was to examine whether performance differences in accuracy could be accounted for by static (CANTAB Emotion Recognition Task) vs. dynamic (Metrisquare Emotion Recognition Task) nature of faces used in these tasks within the same population, which has not been previously examined. Additionally, we chose these tasks because of their similar length and trial numbers as to not have task length/trial number be confounding if performance differences are detected. We have now included this information in the Current Study section to justify task selection.
Reviewer 2 Report
Comments and Suggestions for Authors
I have reviewed the manuscript “Emotion recognition and self-reported emotion processing in 3 alcohol and cannabis co-using young adults” under consideration for publication in Behavioral Sciences. This manuscript was concise and clearly written. Using a matched cohort design, this study compared young adults who co-use alcohol and cannabis (ACC) to healthy controls (HC) on a variety of socio-emotional tasks. Findings suggest no significant differences in emotion recognition, but some impairment in self-reported socio-emotional processing and alexithymia in ACC compared to HC. Strengths include data collection using objective and self-report measures, rigorous screening procedures, and use of matched samples for comparison. I have a few suggestions for improving the manuscript/reporting of this study, which I have attempted to group by the most relevant manuscript section.
Results
1. Although not an explicit aim of the study, it would be interesting to report the correlations between the emotion recognition task performance (i.e., C-ERT/M-ERT) and self-reported functioning (i.e., SEQ/PAQ scores). As the authors state in section 1.6, “affect recognition is critical to healthy socio-emotional functioning”, this might be of interest to readers. Unfortunately, the small sample sizes likely preclude being able to test whether these relationships differ by group, but the authors might consider reporting on the overall relationships.
2. May wish to report relationship between overall C-ERT and M-ERT accuracy and CUDIT/AUDIT scores among the ACC group, similar to what report for the self-report measures.
Discussion
3. It would be helpful to address potential implications/explanations for the differing results based on emotion recognition versus alexithymia/social-emotional functioning. In other words, what does it mean that participants self-report poorer socio-emotional performance, but this was not seen in objective tasks? Relatedly, reporting the relationships among these tasks as suggested above, may contribute to this discussion.
4. While the authors indicate they originally hoped to recruit an alcohol-only comparison group, the use of use healthy controls merits a bit more discussion in terms of implications of the study results. For one, this limits the ability of the current study to address effects of co-use (e.g., are they additive, multiplicative). Second, it confounds 2 noted limitations of the existing literature: 1) lack of study of co-use and 2) lack of non-clinical samples. It would be helpful to address this further.
Author Response
I have reviewed the manuscript “Emotion recognition and self-reported emotion processing in 3 alcohol and cannabis co-using young adults” under consideration for publication in Behavioral Sciences. This manuscript was concise and clearly written. Using a matched cohort design, this study compared young adults who co-use alcohol and cannabis (ACC) to healthy controls (HC) on a variety of socio-emotional tasks. Findings suggest no significant differences in emotion recognition, but some impairment in self-reported socio-emotional processing and alexithymia in ACC compared to HC. Strengths include data collection using objective and self-report measures, rigorous screening procedures, and use of matched samples for comparison. I have a few suggestions for improving the manuscript/reporting of this study, which I have attempted to group by the most relevant manuscript section.
We thank the reviewer for the positive comments on the manuscript and have responded to the suggestions for improving the manuscript below.
Results
- Although not an explicit aim of the study, it would be interesting to report the correlations between the emotion recognition task performance (i.e., C-ERT/M-ERT) and self-reported functioning (i.e., SEQ/PAQ scores). As the authors state in section 1.6, “affect recognition is critical to healthy socio-emotional functioning”, this might be of interest to readers. Unfortunately, the small sample sizes likely preclude being able to test whether these relationships differ by group, but the authors might consider reporting on the overall relationships.
We agree with the reviewer and have conducted Pearson’s correlations across the total sample examining the relationships among these variables and have updated the Methods, Results (section 3.4), and Discussion section to incorporate these new findings.
- May wish to report relationship between overall C-ERT and M-ERT accuracy and CUDIT/AUDIT scores among the ACC group, similar to what report for the self-report measures.
We have now examined Pearson’s correlations between the total C-ERT and M-ERT scores with the CUDIT/AUDIT scores among the ACC group and have updated the Methods, Results (section 3.4), and Discussion section to incorporate these new findings.
Discussion
- It would be helpful to address potential implications/explanations for the differing results based on emotion recognition versus alexithymia/social-emotional functioning. In other words, what does it mean that participants self-report poorer socio-emotional performance, but this was not seen in objective tasks? Relatedly, reporting the relationships among these tasks as suggested above, may contribute to this discussion.
We agree with the reviewer that this was an interesting finding and have integrated literature suggesting there are often discrepancies between self-report and behavioral measurements. Specifically, researchers have argued that higher reliability of self-report vs. performance-based measures (that have greater error variance), more general responses in self-report vs. more specific responses in behavioral tasks, and one-time sampling of these variables (Dang et al., 2020; Carver & Scheier, 1981) could explain the dissociation between self-report and behavioral findings. This is also reflected in the lack of significant correlations between the subjective and objective measures, noted in the manuscript. We have revised the Discussion to integrate this interpretation and have commented on the implications of self-reported difficulties in emotion processing.
- While the authors indicate they originally hoped to recruit an alcohol-only comparison group, the use of use healthy controls merits a bit more discussion in terms of implications of the study results. For one, this limits the ability of the current study to address effects of co-use (e.g., are they additive, multiplicative). Second, it confounds 2 noted limitations of the existing literature: 1) lack of study of co-use and 2) lack of non-clinical samples. It would be helpful to address this further.
The reviewer points out an important limitation of the current study that warrants further Discussion. While, ideally a third group of alcohol-only participants could have been recruited, difficulty finding participants who met criteria for that sample also suggests that frequent co-use of alcohol and cannabis may becoming more prevalent among young adults, in line with some evidence related to changes in cannabis policy effects (Paschall et al., 2022). We now comment on the inability to parse out substance-specific effects when comparing findings to a healthy control group, but also highlight the potential for greater ecological validity of a co-use sample. We also comment that it is unclear whether the current lack of significant behavioral differences are related to the non-clinical sample (with less substance use frequency history, and/or problems) or the co-use of alcohol and cannabis. It is possible that alcohol and cannabis co-use may have somewhat opposing effects on brain and behavior that could counteract each other (as has been suggested in other literature, e.g. Medina et al., 2007; Infante et al., 2018) and resulted in the lack of group differences in behavioral performance. We have now included further information in the Discussion on the implications of the co-use sample for interpreting the current findings.
Round 2
Reviewer 1 Report
Comments and Suggestions for Authors
Overall, I commend the authors for their significant efforts in revising the paper. The improvements are evident in multiple areas, making the research more robust. However, I still have some questions that need clarification, as outlined below:
1. The inclusion criteria for the healthy control (HC) group specify that participants should have "1) ≤10 lifetime uses of cannabis, 2) no cannabis use in the past month, and 3) no binge drinking history." Could the authors clarify the rationale behind these specific criteria? Why were non-users of cannabis and alcohol, or those with no history of drinking in the past month, not chosen as the standard for the control group? It would be helpful to understand the theoretical basis for defining non-binge drinkers as healthy controls, especially when studying a population with known substance use. Further context about how this aligns with existing research or clinical definitions of "healthy" would be appreciated.
2. Could the authors clarify the sequence in which the two emotion recognition tasks were presented to participants? Did participants complete these tasks in a specific order, or were they presented randomly or counterbalanced? The task sequence is crucial, as it might affect the study's outcomes due to factors like task fatigue, learning effects, or order bias.
Author Response
Overall, I commend the authors for their significant efforts in revising the paper. The improvements are evident in multiple areas, making the research more robust. However, I still have some questions that need clarification, as outlined below:
We thank the reviewer for the positive comments regarding the revised manuscript and have addressed the additional recommendations by the reviewer within the manuscript and in our comments below.
- The inclusion criteria for the healthy control (HC) group specify that participants should have "1) ≤10 lifetime uses of cannabis, 2) no cannabis use in the past month, and 3) no binge drinking history." Could the authors clarify the rationale behind these specific criteria? Why were non-users of cannabis and alcohol, or those with no history of drinking in the past month, not chosen as the standard for the control group? It would be helpful to understand the theoretical basis for defining non-binge drinkers as healthy controls, especially when studying a population with known substance use. Further context about how this aligns with existing research or clinical definitions of "healthy" would be appreciated.
We thank the reviewer for the opportunity to clarify our decision regarding the exclusionary criteria for the healthy control group. Given the high prevalence of any lifetime cannabis or alcohol use by young adulthood (Patrick et al., 2023), we allowed some low levels of cannabis and alcohol use in the current control sample for greater ecological validity and generalizability of the study’s findings. The cannabis use criteria for controls are similar to referenced studies, including Platt et al., 2010 (no more than 10 lifetime cannabis uses for healthy controls), Bayrakci et al., 2015 (cannabis use allowed for healthy controls if no cannabis dependence diagnosis), and Hindocha et al., 2014 (control participants were eligible if they had never smoked cannabis more than twice in any month of their lifetime). Additionally, given the high prevalence (>67%) of any past month alcohol use in young adults (Patrick et al., 2023), we excluded specifically for binge-drinking in the control group given prior research suggesting that binge-drinking may be associated with emotion processing impairments (e.g. Lannoy et al., 2018). We have updated the Methods section to include this justification for the exclusionary criteria.
- Could the authors clarify the sequence in which the two emotion recognition tasks were presented to participants? Did participants complete these tasks in a specific order, or were they presented randomly or counterbalanced? The task sequence is crucial, as it might affect the study's outcomes due to factors like task fatigue, learning effects, or order bias.
We agree with the reviewer that this is important to clarify in the manuscript. The two emotion recognition tasks were completed in the same order by all participants, such that the Metrisquare Emotion Recognition Task was completed first, followed by questionnaires (not reported in the current study), after which participants completed the CANTAB Emotion Recognition Task. While within-subjects designs are sensitive to carryover effects and subject to order bias, this concern was alleviated for the current between-subjects design. The fixed order of task presentation is in line with other research utilizing two emotion processing tasks in a between-groups design (Leganes-Fonteneau et al., 2020). Importantly, the two emotion recognition tasks did not directly follow each other but were separated by several questionnaires that participants completed to reduce task fatigue. Given the distinct nature of the facial stimuli used in each task, and no within-subjects aims of the current study, carryover effects were not expected. The Methods section has been updated to include information and justification on task presentation. We have also switched the order of how the tasks are presented in the Methods and Results sections to reflect their administration order, and Figure 1 has been updated to reflect this change. Additionally, we discuss the benefits of future research on this topic utilizing counterbalanced presentation of tasks in within-subjects studies and longitudinal assessments in the Discussion section.